# Evolution of corneal transplantation techniques and their indications in a French corneal transplant unit in 2000–2020

Vianney Malleron[1], Florian Bloch[1], Yinka Zevering[1], Jean-Charles Vermion[1], Axelle Semler-Collery[1], Christophe Goetz[2], Jean-Marc Perone[3]*

1 Department of Ophthalmology, Metz-Thionville Regional Hospital Center, Mercy Hospital, Metz, France, 2 Research Support Unit, Mercy Hospital, Metz-Thionville Regional Hospital Center, Metz, France, 3 Institut Jean Lamour, Lorraine University, Nancy, France

* jm.perone@chr-metz-thionville.fr

## Abstract

**Data Availability Statement:** The data are publicly available on doi 10.5281/zenodo.5913083.

**Funding:** The author(s) received no specific funding for this work.

### Purpose

This retrospective cohort study assessed the evolution of corneal transplantation and its indications in the last 21 years (2000–2020) in a specialized ophthalmology department in a tertiary referral center in France.

### Methods

The surgical techniques and indications, patient age and sex, and postoperative best-corrected visual acuity (BCVA) 6 months after keratoplasty were extracted.

### Results

In total, 1042 eyes underwent keratoplasty in 2000–2020. Annual numbers of corneal transplantations increased by 2.2-fold. Penetrating keratoplasty (PKP) was the sole technique for the first 11 years. Descemet stripping automated endothelial keratoplasty (DSAEK) and Descemet membrane endothelial keratoplasty (DMEK) were introduced in 2011 and 2014, respectively. Cases of both quickly increased, accounting for 28% and 41% of cases in 2015–2020, respectively. Eventually, DSAEK and DMEK were respectively used for most pseudophakic bullous keratopathy (PBK) and all Fuchs endothelial cell dystrophy (FECD) cases. PKP cases declined to 27%. Deep anterior lamellar keratoplasty (DALK) was rare (3% of all cases). These changes associated with rises in PBK and particularly FECD cases, and a strong decline in keratoconus, causing FECD, PBK, and keratoconus to move from being the 4th, 1st, and 3rd most common indications to the 1st, 2nd, and 6th, respectively. On average, BCVA improved by 0.1–0.3 logMAR. Patient age dropped steadily over time. Female predominance was observed.

**Competing interests:** The authors have declared that no competing interests exist.

## Conclusions

The invention of DSAEK and then DMEK precipitated an enormous change in clinical practice and a large expansion of keratoplasty to new indications. This study confirms and extends previous findings in other countries.

## Introduction

Corneal transplantation is currently the most widely performed tissue transplant procedure in the world: in 2019, 5,032 and 84,921 people in France and the USA received a corneal transplant, respectively [1, 2]. This surgery has a history spanning several centuries but first became clinically useful in 1905, when Eduard Konrad Zirm in Olomouc in Central Europe conducted the first successful full-thickness human corneal transplantation [3]. For much of the 20[th] century thereafter, Zirm's technique (denoted penetrating keratoplasty [PKP]) remained the treatment of choice for a wide range of corneal disorders that affected one or more of the corneal layers.

Later in the 20[th] century, the prominent clinical role of PKP began to be challenged by developments in lamellar keratoplasty techniques, where only the diseased corneal layer is replaced by a graft. While this concept was first forwarded in 1888 [4] and revived in 1956 by a successful case of posterior lamellar endothelial keratoplasty [5], it did not become a viable clinical option until the 1970s, when Anwar proposed deep anterior lamellar keratoplasty (DALK) for keratoconus. This technique involves completely or partially replacing the corneal stroma of the patient while preserving the healthy endothelium [6]. At present, the most commonly used DALK technique is the one that was published by Anwar and Teichmann in 2002, where the stroma is dissected and separated by injecting an air bubble between the pathological stroma and the healthy Descemet membrane [6].

The next major change in the lamellar keratoplasty field came around the same time as the publication of the air bubble DALK technique: in 1998–2000, Melles proposed a method for replacing the diseased endothelium in corneal diseases such as bullous keratopathy and Fuchs endothelial corneal dystrophy (FECD). This method was termed deep lamellar endothelial keratoplasty (DLEK) and involves (i) dissecting a posterior lamellar disc from the recipient cornea, (ii) inserting a folded donor corneal disc consisting of posterior stroma, Descemet membrane, and endothelium into a self-healing 5-mm tunnel incision, (iii) unfolding the disc, and (iv) then securing it against the recipient tissue with an air bubble [7–9]. However, while DLEK surpassed PKP in terms of outcomes and complications, it remained a challenging technique [10, 11]. Therefore, in 2004, Melles proposed that the diseased Descemet membrane and its endothelium could be selectively removed with a reversed Sinskey hook (a procedure known as descemetorhexis), after which the folded graft used in DLEK would be introduced into the anterior chamber [12]. Gorovoy and colleagues then simplified this procedure by introducing an automated microkeratome that dissected the donor posterior lamella in a standardized manner [13]. This so-called Descemet stripping automated endothelial keratoplasty (DSAEK) procedure rapidly started replacing PKP worldwide because its lower invasiveness and complication rates meant that it could be offered to the patient before the disease had advanced; this resulted in significantly superior outcomes [14]. Around this time, however, Melles refined the procedure further by only transplanting Descemet membrane and endothelium [15]. This Descemet membrane endothelial keratoplasty (DMEK) technique yields superb visual outcomes compared even to DSAEK and allows the remaining anterior layers of

the donor cornea to be used for DALK [16, 17]. While DMEK is a more complex procedure that associates with higher rates of postoperative graft dehiscence, surgeon experience can overcome this issue [18, 19].

Several studies suggest that these technical developments have led to significant changes globally in the use of keratoplasty techniques and their indications (S1 and S2 Tables) [2, 20–37]. For example, when Tan et al. analyzed their 2002–2011 series, they found that the introduction of DSAEK into clinical practice in 2007 associated with a marked drop in PKP surgeries and FECD moving from the third most common keratoplasty indication to the first [30]. However, few studies have examined the changes after 2015. In addition, the impact of DMEK on clinical practices and indications has not been clearly described to date. Moreover, there is little information about current practices in France.

The present retrospective study analyzed the cases of all consecutive patients who underwent surgery between 2000 and 2020 in a French corneal transplantation unit to elucidate how the inventions of DALK, DSAEK, and DMEK have affected the surgical landscape, the main indications for corneal transplantation, and the surgical outcomes.

## Patients and methods

### Study design and ethics

This retrospective single-center study was performed in the Regional Metz-Thionville Hospital Center, Grand Est, France. All procedures were conducted in accordance with the principles of the Declaration of Helsinki. The study was approved by the Ethics Committee of the French Society of Ophthalmology (IRB 00008855 Société Française d'Ophtalmologie IRB#1). The need for explicit consent from the patients to use their anonymized surgery-related data was waived due to the retrospective nature of the study.

### Patient selection and data collection

The study cohort consisted of all consecutive patients who underwent corneal transplantation in a tertiary referral center (Regional Metz-Thionville Hospital Center) between January 2000 and December 2020. All surgeries were conducted by a single surgeon (JMP). The indications for surgery, the surgical techniques that were used, patient age, sex, and the best-corrected postoperative visual acuity (BCVA) 6 months after keratoplasty were extracted retrospectively from the medical records and the French corneal transplantation waiting list (Agence Française de Biomédicine). BCVA data were only available from 2010 onwards because that was when they started to be routinely and prospectively recorded in the medical database. FECD was diagnosed on the basis of clinical signs and specular microscopy. Bullous keratopathy was diagnosed when pseudophakic or aphakic patients exhibited corneal endothelial decompensation, often due to underlying FECD. Viral, bacterial, parasitic, and fungal infectious pathologies were diagnosed on the basis of the clinical signs and microbiological cultures. When an eye underwent another cornea transplant due to failure of the previous graft, the indication was surgical revision; the original indication was no longer applied.

### Statistical analyses

The data were recorded prospectively and were complete for all patients. Continuous and categorical data were expressed as mean±standard deviation or *n* (%), respectively. Trends in the collected variables over time were analyzed by using the Mann-Kendall Trend test. All statistical analyses were performed with SAS software (version 9.3, SAS Inst., Cary, NC, USA). The significance threshold was set at 5%.

## Results

Between January 2000 and December 2020, 1084 eyes underwent corneal transplantation in our hospital. Of these, the vast majority ($n = 1042$; 96%) were conducted for pseudophakic bullous keratopathy (PBK), FECD, keratoconus, trauma, infectious pathologies, and regrafts after graft failure. The remaining ($n = 42$) were conducted for neurotrophic keratitis, corneal congenital anomaly, and stromal dystrophy ($n$ ranging from 3 to 18). The latter cases were removed from the database to help identify major temporal changes in clinical practice. Thus, the study cohort consisted of 1042 eyes.

### The predominant technique has changed markedly over time from PKP to DMEK

On average, 49±23 (range, 15–97) transplants were performed annually. The annual number of corneal transplantations increased by 2.2-fold between 2000–2010 and 2011–2020 (Fig 1).

Four techniques were used: in order of the most to least frequently used over the 21 years, they were PKP (60% of 1042 surgeries), DSAEK (19%), DMEK (18%), and DALK (3%). Other techniques were not used. Apart from a few DALK cases in 2006–2007, PKP was the only technique that was used for the first 11 years. In 2011, the first cases of DSAEK ($n = 2$) were conducted. This was followed by a large and enduring increase in both DSAEK cases and total cases. In 2014, the first cases of DMEK ($n = 3$) were performed. The numbers of annual DMEK cases then grew so strongly that two years later (2016), DMEK was the most commonly performed technique (46% *vs*. 27% and 26% for DSAEK and PKP, respectively). DMEK case numbers remained relatively constant from 2016 onwards. Since the numbers and frequencies of annual DSAEK cases remained relatively steady, the growth in DMEK cases tended to come at the expense of PKP cases, which dropped from ~100% in 2000–2010 to 27% in 2020. This drop was significant, as determined by a Mann-Kendall Trend test ($p = 0.042$) (Fig 1). DALK cases remained infrequent throughout (Fig 1).

### The predominant indication has changed markedly over time from PBK to FECD

In order from the most to the least common, the keratoplasties over the 21 years were conducted for PBK (31%), FECD (24%), regraft (21%), keratoconus (11%), infectious pathologies (8%), and trauma (5%).

Until 2013, PBK was the main indication for corneal transplantation (42%/year), followed by keratoconus (16%), regraft (15%), FECD (10%), infections (10%), and trauma (7%). This changed in 2013–2016: regraft initially became the most predominant indication but then was strongly overtaken by FECD. PBK eventually became the second most predominant indication, followed by regraft. In 2016–2020, FECD, PBK, and regraft accounted for 43%, 24%, and 21% of cases per year, respectively. Around this time, there was also a marked decrease in keratoconus cases to 3% in 2016–2020, making it now the least common indication. Trauma and infection cases did not show substantial changes over time (Fig 2).

### Treatment for PBK switched largely from PKP to DSAEK

All PBK cases were treated with PKP until 2011, when one case was treated with DSAEK. From then on, the vast majority of PBK cases were treated with DSAEK. At that timepoint, there was also an enduring rise in PBK cases from on average 13/year in 2000–2011 to 19/year in 2012–2020. In 2014, a PBK case was treated with DMEK for the first time. However, while

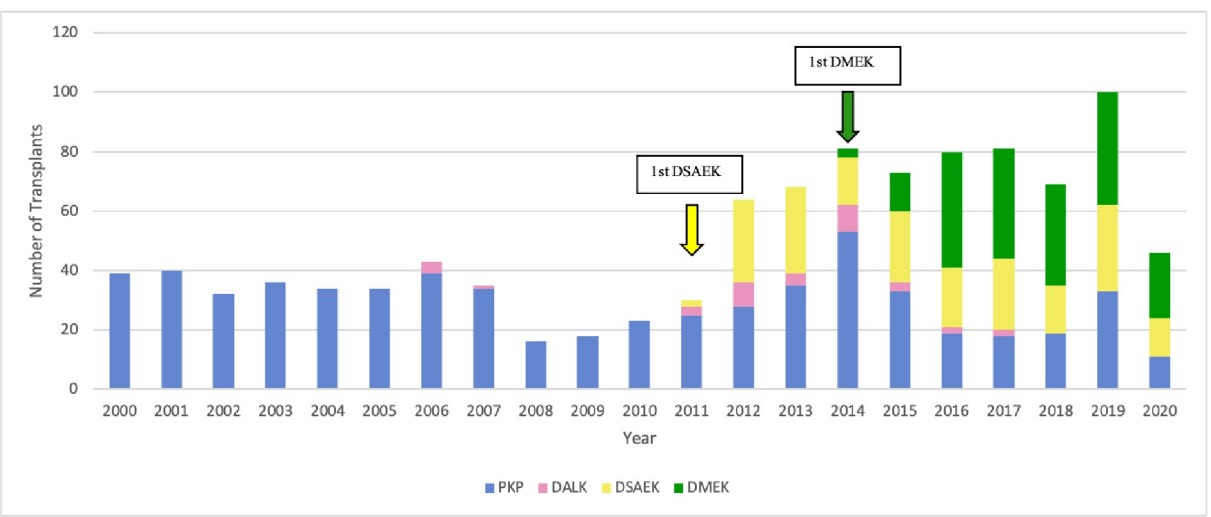

**Fig 1. Changes in surgical techniques over time.** DALK, deep anterior lamellar keratoplasty; DSAEK, Descemet stripping automated endothelial keratoplasty; DMEK, Descemet membrane endothelial keratoplasty; PKP, penetrating keratoplasty.

DMEK was employed for PBK occasionally in later years, DSAEK remained the mainstay of treatment for this indication (Fig 3).

## FECD cases were initially rare and treated with PKP but are now very common and treated with DMEK

Until 2010, only a handful of FECD cases were treated with keratoplasty every year (range, 1–7 cases/year), and all were treated with PKP. In 2011, the first FECD case was treated with DSAEK. DSAEK then became the preferred treatment for the next three years. Around this

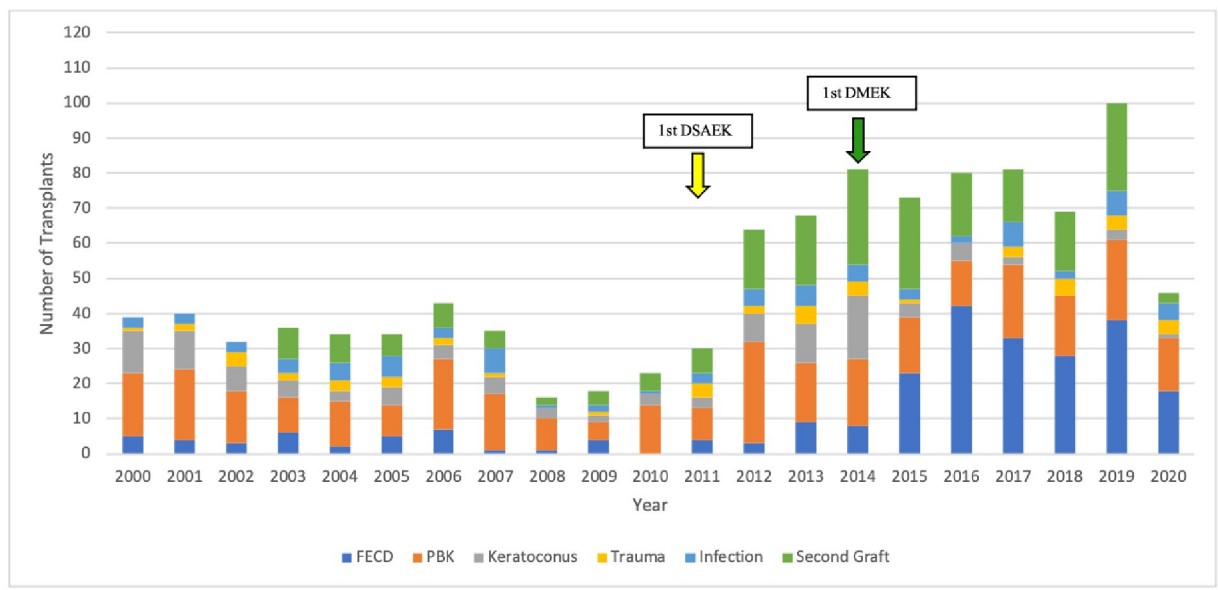

**Fig 2. Changes over time in the indications for corneal grafting.** DSAEK, Descemet stripping automated endothelial keratoplasty; DMEK, Descemet membrane endothelial keratoplasty; FECD, Fuchs endothelial cell dystrophy; PBK, pseudophakic bullous keratopathy.

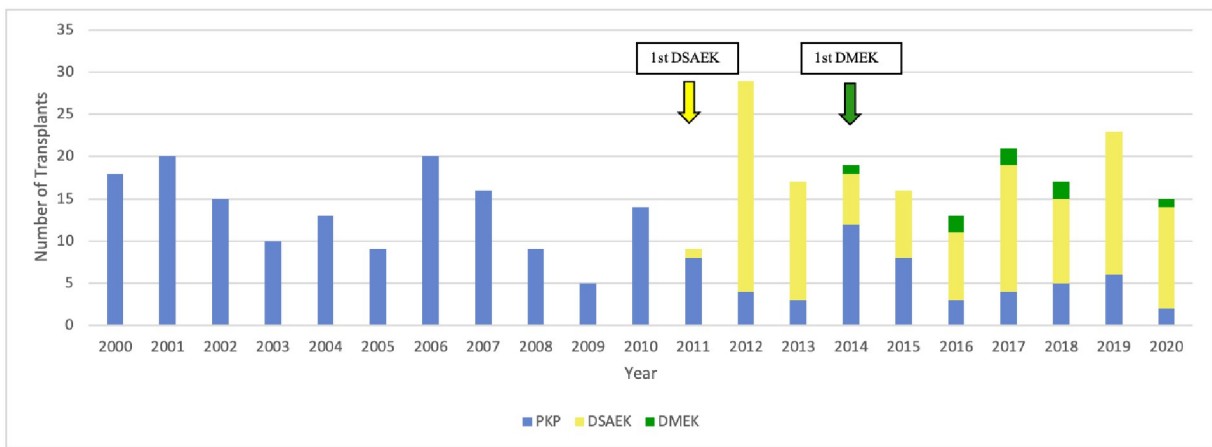

**Fig 3. Changes over time in the type of surgery used for PBK.** DSAEK, Descemet stripping automated endothelial keratoplasty; DMEK, Descemet membrane endothelial keratoplasty; PKP, penetrating keratoplasty.

time, the annual FECD cases started rising. In 2014, the first FECD case was treated with DMEK. Thereafter, DMEK quickly became the preferred treatment for FECD and DSAEK fell out of favor. These changes are associated with an increase in the number of FECD cases (up to 44 annually) (Fig 4). As a result, FECD cases went from being the fourth to the first most common indication (Fig 2).

## Keratoconus cases were increasingly being treated with DALK but have now become rare

Until 2015, there were seven cases of keratoconus per year on average. All cases were treated with PKP until 2006, when five cases were treated with DALK. Thereafter, PKP remained the treatment of choice until 2012, at which point keratoconus started to be increasingly treated with DALK and keratoconus cases started rising. However, starting in 2015, the number of keratoconus cases that underwent keratoplasty dropped sharply to an average of 2 per year (Fig 5).

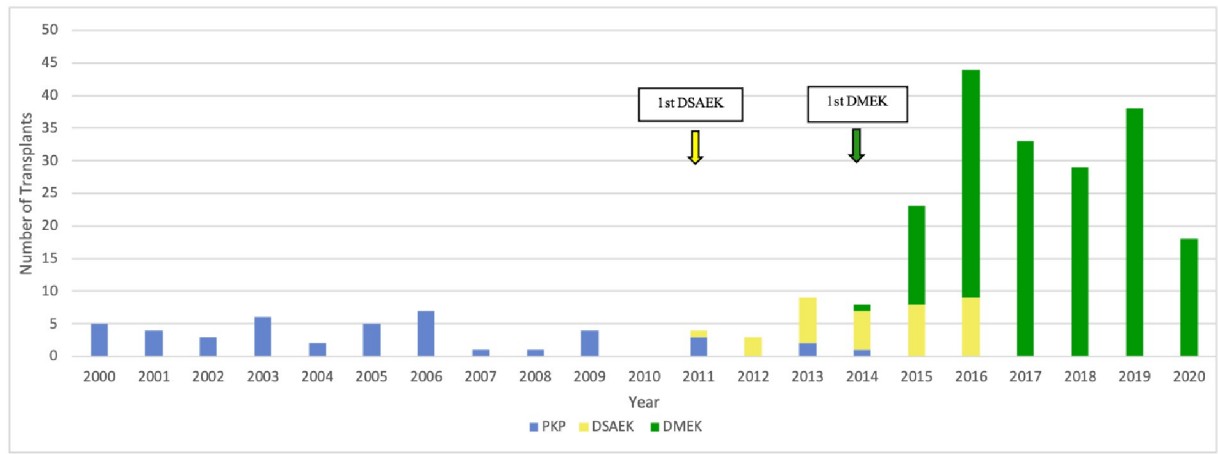

**Fig 4. Changes over time in the type of surgery used for FECD.** DSAEK, Descemet stripping automated endothelial keratoplasty; DMEK, Descemet membrane endothelial keratoplasty; FECD, Fuchs endothelial cell dystrophy; PKP, penetrating keratoplasty.

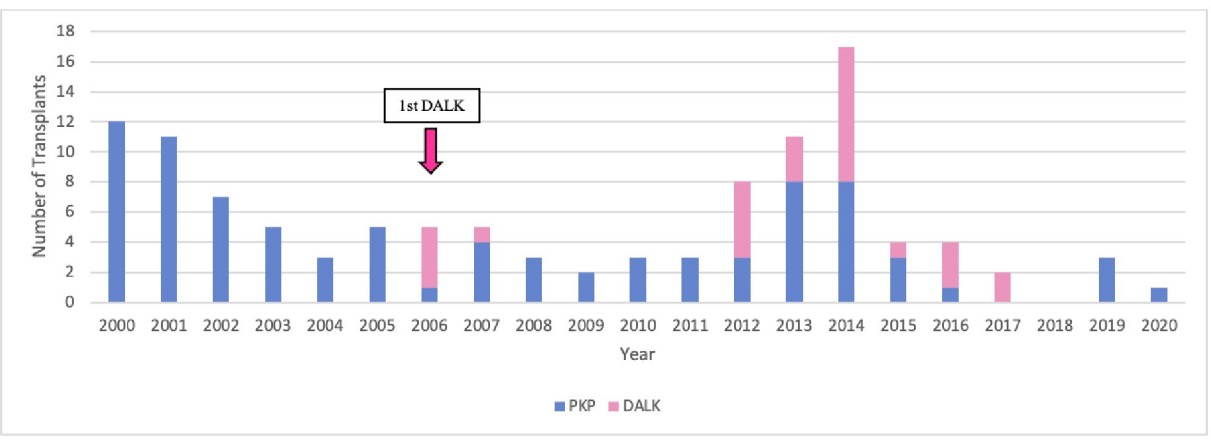

**Fig 5. Changes over time in the type of surgery used for keratoconus.** DALK, deep anterior lamellar keratoplasty; PKP, penetrating keratoplasty.

## Regraft cases have risen sharply and are increasingly being treated with DSAEK or DMEK

Over the 21 years, 221 keratoplasties were conducted for failure of a first transplant (20%; this includes a 10% total graft rejection and 10% total graft failure rate). The regraft cases started to emerge in 2003 but remained low (1–9/year; mean, 6/year) until 2012. At that point, revision keratoplasties almost tripled. These numbers generally remained high thereafter (mean, 19/year). The increase in regraft cases directly reflects the advent of DSAEK and DMEK. PKP was responsible for nearly all of the graft rejections in the cohort (due to its thick and therefore more immunologically provocative graft) but only 1% of the graft failures. PKP rejections and failures were initially treated with a second PKP until 2012, which was when DSAEK was first used for revision keratoplasty. Thereafter, DSAEK was increasingly used after PKP rejection/failure. By contrast, while the DSAEK and especially DMEK grafts were rarely rejected (due to their thinness), 16% and 11% of these grafts failed, respectively. The DSAEK failures were mostly caused by excessive stromal damage that could not be improved sufficiently with DSAEK. Most were repaired with PKP. The DMEK failures were mostly caused by graft detachment that significantly affected graft function and could not be reversed with rebubbling. These cases were generally reoperated with a second DMEK or a DSAEK. Thus, DSAEK was increasingly performed for regraft; in 2019, it was used for 44% of all revision cases. DMEK was first used for revision keratoplasty in 2014 and its use in regrafts rose modestly over time to 24%. Nonetheless, throughout these changes, PKP remained a favored option for revision cases (Fig 6).

## Average patient age dropped over time

On average, the patients were 74±6 years old. There is a U-curve in the age data: patients were consistently older than 75 years on average until around 2010: at that point, average age started to fall below 75 years to nadirs of 62 and 64 years in 2013 and 2016, respectively. Thereafter, average age climbed to between 70 and 75 years. This pattern partly reflects the use of PKP in increasingly younger patients: until 2010, the average patient undergoing PKP was 79 years old but in the next 11 years, the average patient age was 70 years. This drop in age of PKP cases over time was statistically significant (p = 0.005) (Fig 7).

The increase in whole cohort age that started around 2017 reflects the strongly increasing rates of DSAEK and especially DMEK in this period: both techniques were conducted in older

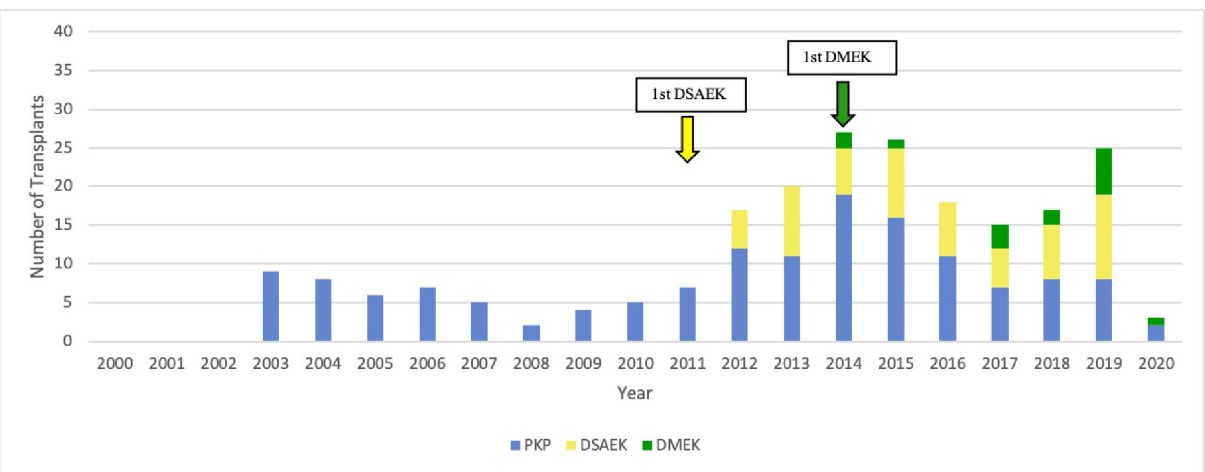

**Fig 6. Changes over time in the type of surgery used for regrafts.** DSAEK, Descemet stripping automated endothelial keratoplasty; DMEK, Descemet membrane endothelial keratoplasty; PKP, penetrating keratoplasty.

patients (average age, 77 years for DSAEK and 75 years for DMEK). However, as with the PKP subgroup, patient age also tended to fall in the DSAEK subgroup (from 83 years in 2011 to 74 years in 2020) and the DMEK subgroup (from 81 years in 2014 to 73 years in 2020) (Fig 7).

The patients who received DALK were the youngest of all patients (47±11 years).

## Women predominated at most timepoints

Women constituted 58% of the total cohort; in 18 of the 21 years (the exceptions were 2000, 2002, and 2014), women formed ≥50% of the yearly cohort. This was also observed for the PKP (mean, 53%) and especially the DSAEK (67%) and DMEK (72%) subgroups (S1–S4 Figs).

## Postoperative BCVA improved over time for all keratoplasty techniques

Fig 8 shows the BCVA at 6 postoperative months in 2010–2020 for all keratoplasty techniques. Postoperative visual acuity improved over time by 0.1–0.3 logMAR on average for all techniques. DMEK associated with the best visual outcomes (mean, 0.16 logMAR), followed by DALK (0.3 logMAR) and DSAEK (0.4 logMAR). PKP had the poorest outcomes (0.6 logMAR) (Fig 8).

## Discussion

The present retrospective study examined how corneal transplantation methods and their indications have changed over the last 21 years in a specialized ophthalmology department in France. Fig 9 summarizes the major changes that have occurred in the field and the department. Thus, up until the beginning of the 21[st] century, PKP was the sole treatment of choice in the field for the predominant indications for keratoplasty then. The publication of air bubble DALK for anterior opacities in 2002 and the invention of first DSAEK for endothelial pathologies in 2004 and then its technically more challenging counterpart DMEK in 2006 has completely revolutionized the field with enormous speed: this is clearly demonstrated by our department's experiences, which are depicted in the present study.

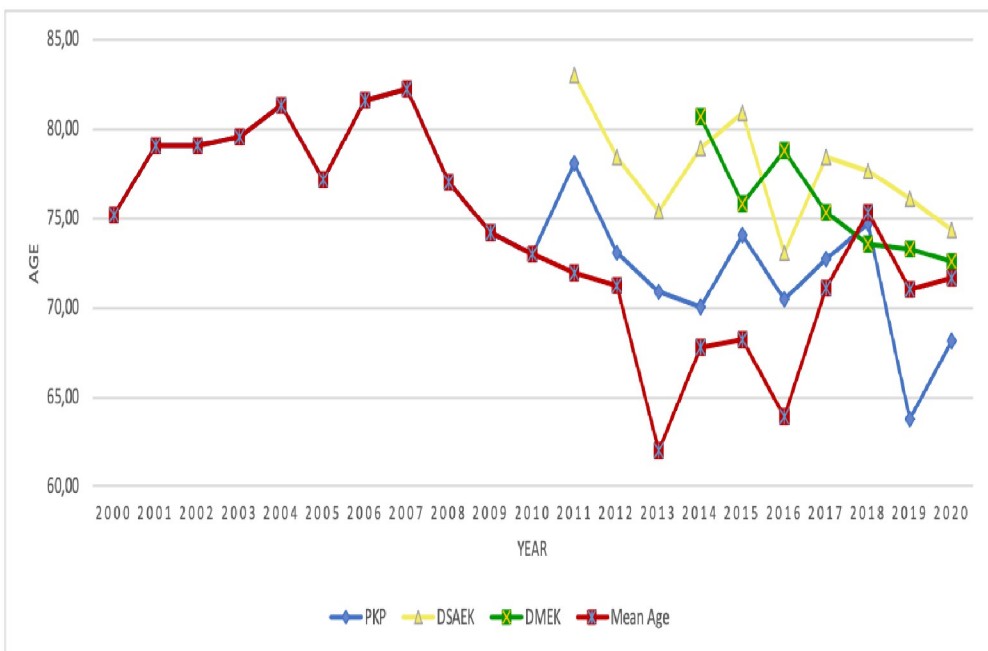

**Fig 7. Changes in average patient age over time in the whole cohort and in patients undergoing PKP, DSAEK, and DMEK.** DSAEK, Descemet stripping automated endothelial keratoplasty; DMEK, Descemet membrane endothelial keratoplasty; PKP, penetrating keratoplasty.

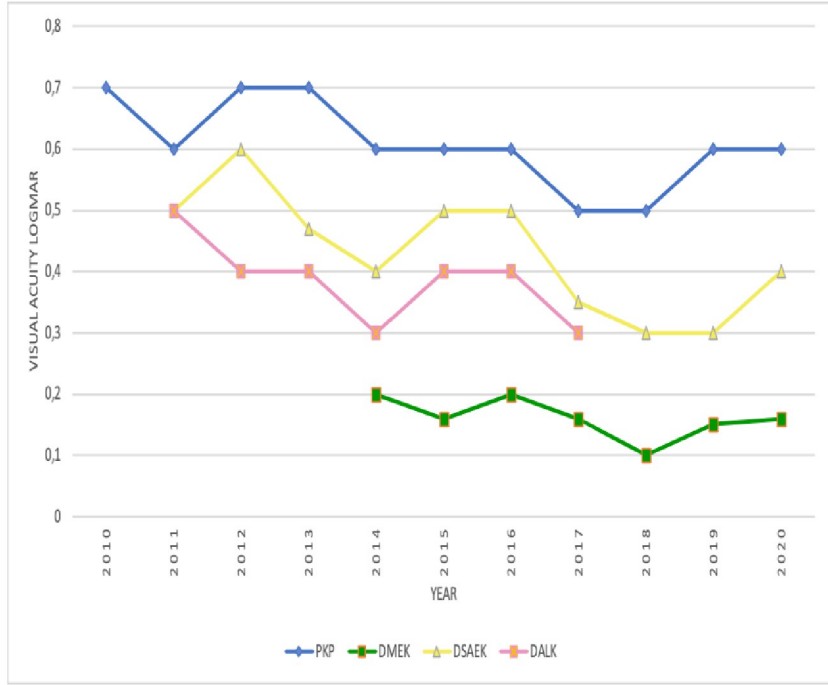

**Fig 8. Changes over time in terms of mean BCVA 6 months after keratoplasty.** BCVA, best corrected visual acuity; DALK, deep anterior lamellar keratoplasty; DSAEK, Descemet stripping automated endothelial keratoplasty; DMEK, Descemet membrane endothelial keratoplasty; PKP, penetrating keratoplasty.

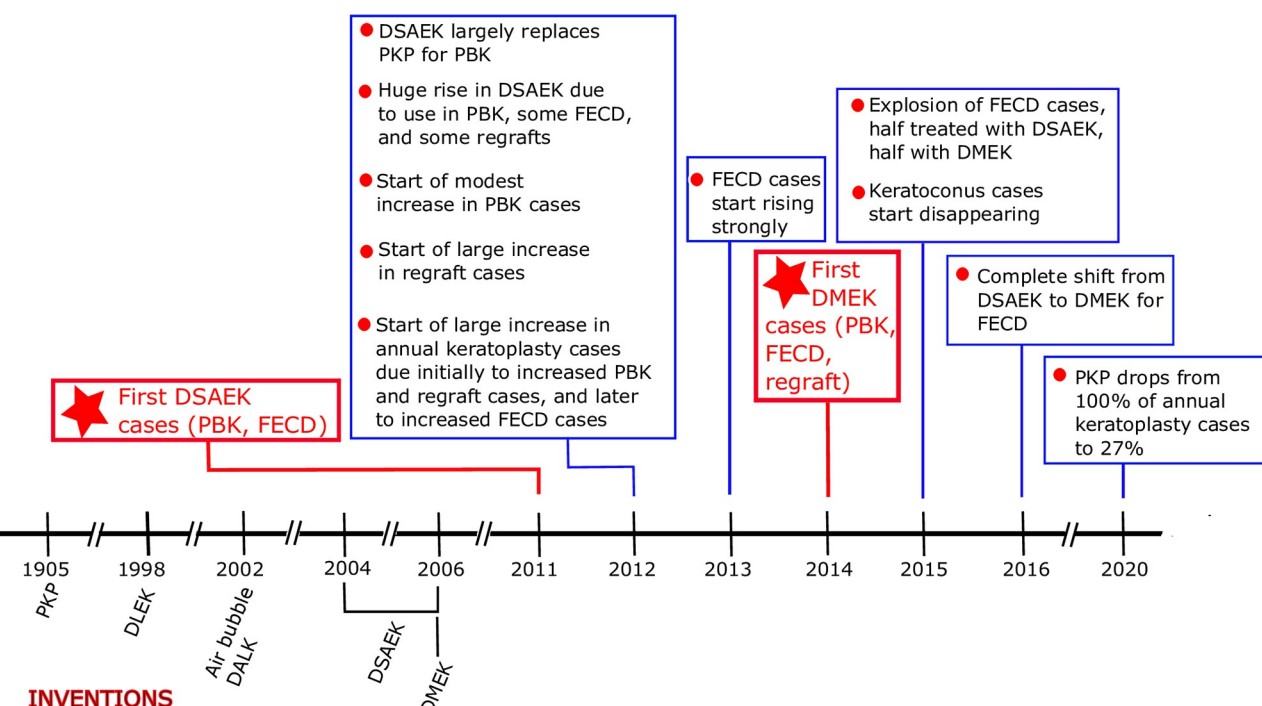

**Fig 9. Timeline showing the inventions of PKP, DALK, DSAEK, and DMEK and their effects on our clinical practice and indications.** DALK, deep anterior lamellar keratoplasty; DSAEK, Descemet stripping automated endothelial keratoplasty; DMEK, Descemet membrane endothelial keratoplasty; FECD, Fuchs endothelial cell dystrophy; PBK, pseudophakic bullous keratopathy; PKP, penetrating keratoplasty.

## Changes in PKP, DSAEK, and DMEK case numbers and indications

In 2011, our department tried DSAEK for the first time in one PBK and one FECD case. The good visual outcomes and shorter recovery durations caused us to switch largely from using PKP for PBK to DSAEK in the following year (2012). The year 2012 was also when annual PBK cases rose by 220%; this increase has largely endured. Thus, the invention of DSAEK expanded the number of PBK cases that could be effectively treated with keratoplasty.

Another marked change started the next year (2013): PKP was largely abandoned in favor of DSAEK for FECD as well. At this point, the total annual FECD cases started rising sharply; most were treated with DSAEK. Then, in the following year, the surgeon performed DMEK for the first time in a few PBK, regraft, and FECD cases. This led to a rapid switch from DSAEK to DMEK for FECD in the following year. By 2017, all FECD cases were being treated with DMEK. This switch to DMEK associated with a huge 3-fold increase in the annual number of FECD cases. As a result, FECD changed from being the fourth most common indication to the first. This associated with a drop in PBK from first to second place. Thus, the invention of DSAEK and then DMEK hugely expanded the number of FECD cases that could be effectively treated with keratoplasty.

These changes were accompanied by a large and sustained rise in regraft cases. This reflects the complications of the DSAEK and DMEK techniques.

Due to the increased numbers of PBK, FECD, and regraft cases, the total annual cases of keratoplasty rose by 2.2-fold from 2000–2010 to 2011–2020; this increase started in 2012, one year after DSAEK was first tried.

During this time, keratoconus dropped from third to last place as a keratoplasty indication. This is partly because of improvements in rigid and scleral lenses but also because of the rise of cross-linking (CXL) treatment, which was first used to manage keratoconus in 2003. CXL involves instilling riboflavin and then irradiating the cornea with UVA [38–40]. This induces covalent bonds between the corneal collagen fibers, thus stiffening the cornea. Since this prevents further progression of keratoconus in 90% of cases, CXL reduces the need for corneal transplantation, which is now generally only conducted in the few cases that evince keratoconus progression [41–46].

Thus, over the space of 6 years, the keratoplasty landscape in our clinical practice changed radically (Fig 9). Rather than PKP being used for every indication and FECD being treated only rarely, PKP has been largely supplanted by especially DMEK (now used for all FECD cases) but also DSAEK (now used for most PBK and some regraft cases), and total annual keratoplasty, FECD, and PBK cases have increased markedly.

## Changes in other countries

Keratoplasty evolution studies and internet reports show that most of the trends we observed have also been found in other countries, although there are some differences (S1 and S2 Tables) [2, 20–37]. These studies are nearly all based on national eye bank or transplant registry data and most cover the periods before, during, and after 2004–2006 (when DSAEK and then DMEK emerged). Study durations range from 10 to 27 years. Half report up to 2009–2012 while all but one of the rest report up to 2014–2017: the most recent data were reported by the Eye Bank Association of America (2005–2019) [2]. All but one relate to industrialized countries: the exception is Iran [23].

**Increase in annual case numbers.**   The literature shows that annual keratoplasty cases increased on average by 1.6-fold after 2005 (cf. 2.2-fold in our study). Most increases ranged from 1.3- to 2.0-fold. The Netherlands showed a large increase in annual cases (3-fold), possibly reflecting the fact that a Dutchman invented both DSAEK and DMEK [7, 8, 12, 15]. A Polish study also recorded a 4.6-fold increase between 1989 and 2014; these changes may reflect legal and infrastructural changes as well as the inventions in lamellar keratoplasty [27]. A French national database study covering the period 2004–2015 reported only a 1.1-fold increase; this may reflect the fact that the study start year was also the year endothelial keratoplasty was introduced in France [24]. Canada reported increases of 1.1-, 1.1-, and 2.6-fold, possibly reflecting regional differences (S1 and S2 Tables) [20, 30, 34].

**Uptake of DSAEK and DMEK.**   Nearly all countries introduced DSAEK by 2008. DMEK entered clinical practice more slowly: while Australia and Germany introduced it in 2008–2009, the USA, Singapore, New Zealand, Iran, and Greece first started using it in 2011–2015. The French national database study does not discriminate between DSAEK and DMEK, so it is not clear when DMEK was first used; however, endothelial keratoplasty started in 2004 (S1 and S2 Tables). In our hospital study, DSAEK and DMEK were introduced in 2011 and 2014, respectively; this apparent lateness reflects the fact that nearly all previous studies were based on eye bank or transplant registry data that registered the earliest attempts at these new techniques nationally. The slower uptake of DMEK internationally probably reflects the fact that this procedure is technically more challenging than DSAEK due to the thinness of the graft and its tendency to roll up and dehisce from the graft bed; consequently, mastering DMEK requires a steeper learning curve [19, 47–49].

In the literature, the uptake of DSAEK at the last study timepoint ranged from 9% to 59% (average, 31%; cf. 27% in our study). In two German studies, DSAEK was almost immediately supplanted by DMEK [21, 29]. None of the studies directly reported DMEK rates. DALK

either started around the time DSAEK and DMEK appeared or was already in use. Like in our series, DALK was rarely conducted in Canada, Germany, and the USA. The UK, New Zealand, and Italy reported an increase in DALK cases from 0% to 12–15% of total keratoplasties while Singapore reported a strong increase from 0% to 44%, and Iran reported a decrease from 17% to 1% (S1 and S2 Tables). These findings are largely consistent with those of Tan et al., who obtained and analyzed the 2011–2012 corneal transplant data from 12 national and two supra-national corneal/organ transplant registries or eye banks. Thus, they observed that endothelial keratoplasty adoption rates were higher in industrialized countries (41–52%) and low in less economically developed countries like Thailand, Philippines, India, and China (1–4%). By contrast, DALK ranged widely from 28% (Singapore, China, and Brazil) to 1% (Philippines) and did not show a distinct difference between industrialized and economically developing countries. This may partly reflect the fact that DALK is still largely dependent on manual skills and less on instrumentation, making it less dependent on the economic status of the country [36]. Moreover, keratoconus rates also vary geographically and some countries have religious strictures against the use of donor material [50, 51]. Low uptake of DALK may also reflect the technical difficulties associated with DALK and the fact that PKP may yield slightly better visual outcomes for keratoconus [52, 53].

**Changes in indications and their treatments.** The international studies also observed that the introduction of DSAEK and DMEK associated with changes in the most common indications, particularly the more recent studies. Thus, FECD went from last to first place in the USA, Germany, the Netherlands, and in the 2004–2015 national registry study in France. In addition, regraft became more common in Canada, New Zealand, Greece, the USA, and the French study. Both PBK and keratoconus dropped in rank over time in nearly all countries (S1 and S2 Tables).

In early studies, before DMEK was introduced, these indication changes associated with a switch from PKP to DSAEK for PBK (average, 63%; range 21–90%; cf. 90% in our study), FECD (70%; 63–90%; cf. 0% in our study), and regraft (27%; 22–30%; cf. 44% in our study). The four studies that were conducted after the introduction of DMEK reported that PKP was abandoned in favor of endothelial keratoplasty for most FECD cases (91%; 70–100%; cf. 100% DMEK in our study); however, it is not clear whether DSAEK or DMEK was performed (S1 and S2 Tables).

## Patient age and sex

We observed that age at keratoplasty dropped over the 21 years from 82 to 62 years of age with a U-curve. This reflects the introduction of DSAEK and especially DMEK, which can and should be offered to younger patients since it has been observed that early management of endothelial dysfunction with DMEK yields better postoperative results compared to waiting for disease progression [54, 55]. Interestingly, we noted that even before we started using DSAEK and DMEK, the PKP patients started getting younger. This may reflect surgeon experience, which led to younger patients being offered PKP.

None of the international studies examined change in average patient age over time, and only an early (1999–2009) British study has comprehensively analyzed the effect of DSAEK and DMEK on patient age and sex. They observed that age had a bimodal distribution, with peaks at 20–40 and 60–80 years, and that this was especially prominent for men and did not change over their study period [31]. Another early study in New Zealand (1990–2009) also observed a similar bimodal age distribution for the whole cohort [32]. However, when this group extended the study time period to 1999–2015, they found that while a bimodal distribution in age was still observed overall, the younger peak dropped over time while the older peak

rose [25]. Moreover, recent Australian and American studies show a unimodal age distribution, with a peak at 70–80 years [2, 28]. When we examined how age distribution changed between 2000–2010 and 2011–2020, we also observed a mild bimodal distribution in the early period (a small peak at 41–50 and a large peak at 91–100) that changed into a unimodal distribution with a peak at 71–80 years (S5 Fig). This change from a bimodal to unimodal age distribution probably reflects the fact that keratoconus (which is predominant in young people) is increasingly not being treated with keratoplasty.

In terms of sex, Canadian, Dutch, and Italian studies showed a female predominance whereas British, Australian, New Zealand, Singaporean, and Iranian studies showed a male predominance. Interestingly, a Greek study observed that after DSAEK was introduced, the male predominance shifted to a female predominance. We also observed a female predominance, especially for DSAEK and DMEK patients, whereas the French registry study, which covered the period up to 2015, observed a male predominance. Thus, a shift to female predominance in keratoplasty patients may be occurring. This trend probably reflects the large increase in cases treated for FECD, which has a strong female predominance and is also a known contributor to the development of PBK [56].

## Ongoing improvements in postoperative BCVA for all techniques

The postoperative visual acuity of our patients who received DMEK was superior to other surgical techniques; it reached 0.15 logmar in 2019 against 0.3 and 0.6 logmar for DSAEK and PKP in 2019, respectively. This has also been observed elsewhere [57–64]. These better postoperative results associate with greater patient satisfaction and better quality of life and make DMEK the technique of choice for managing endothelial dysfunction [65].

## Study limitations

This study has several limitations. First, it is based on retrospective data, which are inherently prone to bias. However, the database from 2010 onwards was maintained prospectively. Second, it is a single center study. However, all lamellar keratoplasties were performed by the same experienced surgeon, which prevented differences between surgeons from confounding the temporal trends.

## Conclusions

The present study shows clearly and comprehensively how the advent of DSAEK and DMEK has created an enduring and expanding revolution in the keratoplasty field, not only in our department but also globally. In fact, the outcomes of these techniques were so convincing that within only a few years, they largely supplanted PKP for PBK and especially FECD. The better outcomes of DMEK compared to DSAEK also mean that DMEK is the method of choice for FECD (especially before the disease decompensates into bullous keratopathy), and cases where the posterior capsule is intact and the anterior segment anatomy is normal. However, DSAEK remains the preferred option for moderate/severe bullous keratopathy and other complex cases where the anterior chamber of the eye is shallow, poorly visualized, or communicates with the vitreous cavity. These changes have also greatly increased the number of patients who are now eligible for eyesight-restoring keratoplasty.

Thus, the inventions of DSAEK and DMEK have enormously advanced the medical field and are greatly improving patient outcomes.

## Supporting information

**S1 Fig. Change in patient sex over time in the whole cohort.**
(DOCX)

**S2 Fig. Change in patient sex over time in PKP.**
(DOCX)

**S3 Fig. Change in patient sex over time in DSAEK.**
(DOCX)

**S4 Fig. Change in patient sex over time in DMEK.**
(DOCX)

**S5 Fig. Change in patient age distribution between 2000–2010 and 2011–2021.**
(DOCX)

**S1 Table. Summary of the literature studying the evolution of keratoplasty techniques and indications before and after DSAEK and DMEK were introduced.**
(DOCX)

**S2 Table. International changes in PKP, DSAEK, and DMEK use and indications.**
(DOCX)

## Author Contributions

**Conceptualization:** Christophe Goetz, Jean-Marc Perone.

**Data curation:** Vianney Malleron, Florian Bloch, Jean-Charles Vermion, Axelle Semler-Collery, Christophe Goetz, Jean-Marc Perone.

**Formal analysis:** Vianney Malleron, Christophe Goetz.

**Funding acquisition:** Christophe Goetz.

**Investigation:** Vianney Malleron, Jean-Marc Perone.

**Methodology:** Vianney Malleron, Christophe Goetz, Jean-Marc Perone.

**Project administration:** Jean-Marc Perone.

**Resources:** Jean-Marc Perone.

**Software:** Christophe Goetz.

**Supervision:** Jean-Marc Perone.

**Validation:** Yinka Zevering, Jean-Marc Perone.

**Visualization:** Vianney Malleron.

**Writing – original draft:** Vianney Malleron.

**Writing – review & editing:** Vianney Malleron, Florian Bloch, Yinka Zevering, Jean-Charles Vermion, Axelle Semler-Collery, Christophe Goetz, Jean-Marc Perone.

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
