## [Decision Letter · Decision Letter 0]

10 Nov 2021

PONE-D-21-29757Evolution of Corneal Transplantation Techniques and Their Indications in a Specialized French Ophthalmology Department in 2000–2020PLOS ONE

Dear Dr. Perone,

Thank you for submitting your manuscript to PLOS ONE. After careful consideration, we feel that it has merit but does not fully meet PLOS ONE’s publication criteria as it currently stands. Therefore, we invite you to submit a revised version of the manuscript that addresses the points raised during the review process.

We look forward to receiving your revised manuscript.

Kind regards,

Timo Eppig

Academic Editor

PLOS ONE

Reviewers' comments:

Reviewer's Responses to Questions

**Comments to the Author**

1. Is the manuscript technically sound, and do the data support the conclusions?

Reviewer #1: Yes

Reviewer #2: Yes

2. Has the statistical analysis been performed appropriately and rigorously? 

Reviewer #1: Yes

Reviewer #2: Yes

3. Have the authors made all data underlying the findings in their manuscript fully available?

Reviewer #1: Yes

Reviewer #2: Yes

4. Is the manuscript presented in an intelligible fashion and written in standard English?

Reviewer #1: Yes

Reviewer #2: Yes

5. Review Comments to the Author

Reviewer #1: Dear editor,

Dear authors,

thank you for inviting me to review this manuscript which summarizes indications and techniques of corneal transplantation in a French ophthalmology department over a period of 21 years.

My comments are as follows:

1) Line 64: typo “Tillett“ and “Barraquer“

2) Line 73: The air bubble is injected to separate Descemet’s membrane from the corneal stroma. Descemet’s membrane is preserved and, therefore, this sentence describing an “air bubble between …. stroma and … endothelium“ should be reworded.

3) Line 78: FECD refers to “Fuchs endothelial corneal dystrophy“. This abbreviation should be used consistently and not redefined.

4) Line 79: typo “of“

5) The introduction summarizes the historical development of keratoplasty techniques. When evaluating different keratoplasty techniques over time, these techniques must of course be presented. Nevertheless, I think that the individual development steps reported in the introduction could be shortened – as this article is not a historical review of keratoplasty techniques.

6) Line 130: Bullous keratopathy was diagnosed based on corneal endothelial decompensation in pseudophakic or aphakic eyes. It therefore may include eyes with underlying FECD (compare line 420).

7) Line 149: Despite excluding these eyes, the authors may would like to specify “other corneal degeneration“.

8) Line 167: “This drop was significant“ compared to what ? Presumingly the number of PKP in 2019 ?

9) As pointed out in comment 6), PBK may then include eyes with endothelial decompensation because of underlying FECD ?

10) Line 174: “Until 2015, PBK was the main indication for corneal transplantation (38%/year),

175 followed by regraft (20%), keratoconus (17%), FECD (10%), infections (10%), and trauma

176 (6%).“

Considering the length of the bars in Figure 2, this is not true: The main indication in 2013 and 2014 was regraft (“second graft“) followed by PBK.

11) Line 176: “This changed in 2015: FECD became the most predominant indication while PBK was

now the second most predominant indication“.

Considering the length of the bars in Figure 2, this is also not true: In 2015, the main indication was again regraft (“second graft“) followed by FECD.

12) The subheadings are successful and point to the next illustration, respectively.

13) Lines 246-251: How was BSCVA assessed – using spectacles or contact lenses ? The abbreviation BSCVA implicates that spectacles were used (“best spectacle corrected visual acuity“), however, this is not stated within the manuscript – even not when introducing the abbreviation BSCVA in lines 31-32 and 126.

14) Line 279: The authors might would like to specify whether the rise of regrafts was attributable to a specific keratoplasty technique.

15) Line 289: I would suggest using “further progression“ instead of “further degeneration“ in the context of corneal crosslinking treatment to stabilize keratoconus.

16) The literature review is well written.

Reviewer #2: The authors present a fascinating article assessing the evolution of types of corneal transplants. The duration of the analysis is 21 years, from 2000 to 2020. The treatments were performed in a clinic specializing in corneal transplants in France. This is a retrospective cohort study. The authors assess indication, surgical techniques, demographic data, and BCVA 6 months after surgery. The total number of patients included in the analysis was 1,042 in the last 20 years.{For the first ten years, only Penetrating grafts have been performed for the 10 first years of analyzing, and then lamellar grafts have started. New procedures have launched in 2011 ( DMEK, DSAEK, DALK). The main indication for keratoplasty changed from Keratoconus to pseudophakic bullous keratopathy (PBK) and all Fuchs

endothelial cell dystrophy (FECD). The proportion of penetrating to lamellar corneal grafts in the last year was 27 to 78 percent. The Authors observed improvement of BCVA in all cases.

In conclusion, the authors state that the introduction of lamellar grafts to corneal surgery was essential in the change in the treatment of corneal disease,

The authors are asked to clarify some issues.

Title line 3 : explain the word,, specialized “ is it Corneal Transplant Unit or Corneal Transplant Center ? And change the title.

Abstract:

Line 43 ,, age dropped steadily over time.” Age of whom?

In the abstract is a lack of numbers ( BCVA before and after),

Line 49 key words : look for more appropriate ones ( indication, surgical technique)

Introduction: line 56 when Eduard Konrad Zirm put ,, in Olomouc in Central Europe. “

Line 72 by Anwar and Teichmann in 2002 insert and published in 2002 y. ( They invented this technique earlier before 2000)

Line 74 dissected separated

Line 75-76 The extensive bubble technique is described above, and DELK should be described here, invented by Gerit Melles.

Line 106 insert Aim of the study:

Line 115 This is unclear to me. The study is retrospective. In that case, did the patient know what he agreed to? Was such consent then necessary?

Ethics Committee of the French Society of Ophthalmology (Approval No. 00008855). Is it a number of approval for this study or all these kinds of studies in France?

Line 124 Put initials of a surgeon ( author of the article?)

Line 128 The date are incomplete BCVA only from 2010 why?

Line 129 Do you have data of the thickness of the cornea ?

Line 251 How many graft rejections occurred in this cohort? How many rebuilding after DSAEK and DMEK occurred?

Line 258 big bubble was not invented in 2002.

Discussion

Line 258 not only CXL

• Please consider to add : 10.1016/j.transproceed.2016.01.056

• Line 350 Please consider to add : 10.3390/jcm10112421

Conclusion

Please add one ending sentence as a conclusion

6. PLOS authors have the option to publish the peer review history of their article (what does this mean?). If published, this will include your full peer review and any attached files.

Reviewer #1: No

---

## [Author Response · Author response to Decision Letter 0]

16 Dec 2021

Point-by-Point Response to reviewers’ Comments

We are grateful for the reviewers for their thorough review of our paper and their thoughtful and helpful comments. We have addressed all points to the best of our ability and believe the revisions have improved our paper. 

Reviewer #1

thank you for inviting me to review this manuscript which summarizes indications and techniques of corneal transplantation in a French ophthalmology department over a period of 21 years.

Reply: Thank you very much for your thorough review of our paper and pertinent comments. We have addressed all points to the best of our ability and believe the revisions have improved our paper. Please note that underlines indicate changed texts in the cited texts below.

My comments are as follows:

1) Line 64: typo “Tillett“ and “Barraquer“

Reply: We deleted these names to address point #5 below.

2) Line 73: The air bubble is injected to separate Descemet’s membrane from the corneal stroma. Descemet’s membrane is preserved and, therefore, this sentence describing an “air bubble between …. stroma and … endothelium“ should be reworded.

Reply: We have rewritten the sentence as follows:

“At present, the most commonly used DALK technique is the one that was published by Anwar and Teichmann in 2002, where the stroma is dissected and separated by injecting an air bubble between the pathological stroma and the healthy Descemet membrane [6].” Page 3, lines 69–72

3) Line 78: FECD refers to “Fuchs endothelial corneal dystrophy“. This abbreviation should be used consistently and not redefined.

Reply: We have made the correction. Page 3, line 76

4) Line 79: typo “of“

Reply: The sentence was unwieldy and was rewritten as follows:

“This method was termed deep lamellar endothelial keratoplasty (DLEK) and involves (i) dissecting a posterior lamellar disc from the recipient cornea, (ii) inserting a folded donor corneal disc consisting of posterior stroma, Descemet membrane, and endothelium into a self-healing 5-mm tunnel incision, (iii) unfolding the disc, and (iv) then securing it against the recipient tissue with an air bubble [7–9].” Pages 3–4, lines 76–80

5) The introduction summarizes the historical development of keratoplasty techniques. When evaluating different keratoplasty techniques over time, these techniques must of course be presented. Nevertheless, I think that the individual development steps reported in the introduction could be shortened – as this article is not a historical review of keratoplasty techniques.

Reply: We removed some historical detail. The paragraph now reads as follows:

“Later in the 20th century, the prominent clinical role of PKP began to be challenged by developments in lamellar keratoplasty techniques, where only the diseased corneal layer is replaced by a graft. While this concept was first forwarded in 1888 [4] and revived in 1956 by a successful case of posterior lamellar endothelial keratoplasty [5], it did not become a viable clinical option until the 1970s, when Anwar proposed deep anterior lamellar keratoplasty (DALK) for keratoconus. This technique involves completely or partially replacing the corneal stroma of the patient while preserving the healthy endothelium [6]. At present, the most commonly used DALK technique is the one that was published by Anwar and Teichmann in 2002, where the stroma is dissected and separated by injecting an air bubble between the pathological stroma and the healthy Descemet membrane [6].” Page 3, lines 63–72

6) Line 130: Bullous keratopathy was diagnosed based on corneal endothelial decompensation in pseudophakic or aphakic eyes. It therefore may include eyes with underlying FECD (compare line 420).

Reply: Indeed, many patients with pseudophakic bullous keratopathy (at least 30–50%) had underlying FECD. This point was added:

“Bullous keratopathy was diagnosed when pseudophakic or aphakic patients exhibited corneal endothelial decompensation, often due to underlying FECD.” Pages 5–6, lines 128–130

7) Line 149: Despite excluding these eyes, the authors may would like to specify “other corneal degeneration“.

Reply: We decided to remove the text “other corneal degeneration” since it meant “corneal stromal dystrophies”. The text now reads:

“The remaining (n=42) were conducted for neurotrophic keratitis, corneal congenital anomaly, and stromal dystrophy (n ranging from 3 to 18).” Page 6, lines 148–149 

8) Line 167: “This drop was significant“ compared to what ? Presumingly the number of PKP in 2019 ?

Reply: A Mann-Kendall Trend test was conducted to determine whether the drop in PKP cases over the 21 years was statistically significant. The text was changed as follows to make this clear:

“Since the numbers and frequencies of annual DSAEK cases remained relatively steady, the growth in DMEK cases tended to come at the expense of PKP cases, which dropped from ~100% in 2000–2010 to 27% in 2020. This drop was significant, as determined by a Mann-Kendall Trend test (p=0.042) (Figure 1).” Page 7, lines 165–168

9) As pointed out in comment 6), PBK may then include eyes with endothelial decompensation because of underlying FECD ?

Reply: Yes, 30–50% of PBK patients had underlying FECD.

10) Line 174: “Until 2015, PBK was the main indication for corneal transplantation (38%/year),

175 followed by regraft (20%), keratoconus (17%), FECD (10%), infections (10%), and trauma

176 (6%).“

Considering the length of the bars in Figure 2, this is not true: The main indication in 2013 and 2014 was regraft (“second graft“) followed by PBK.

11) Line 176: “This changed in 2015: FECD became the most predominant indication while PBK was now the second most predominant indication“.

Considering the length of the bars in Figure 2, this is also not true: In 2015, the main indication was again regraft (“second graft“) followed by FECD.

Reply to #10 & 11: Yes, indeed, these texts are incorrect (the text related to an earlier draft of the figure). They have been corrected as follows:

“Until 2013, PBK was the main indication for corneal transplantation (42%/year), followed by keratoconus (16%), regraft (15%), FECD (10%), infections (10%), and trauma (7%). This changed in 2013–2016: regraft initially became the most predominant indication but then was strongly overtaken by FECD. PBK eventually became the second most predominant indication, followed by regraft. In 2016–2020, FECD, PBK, and regraft accounted for 43%, 24%, and 21% of cases per year, respectively. Around this time, there was also a marked decrease in keratoconus cases to 3% in 2016–2020, making it now the least common indication. Trauma and infection cases did not show substantial changes over time (Figure 2).” Page 7–8, lines 175–182

12) The subheadings are successful and point to the next illustration, respectively.

Reply: Thank you.

13) Lines 246-251: How was BSCVA assessed – using spectacles or contact lenses ? The abbreviation BSCVA implicates that spectacles were used (“best spectacle corrected visual acuity“), however, this is not stated within the manuscript – even not when introducing the abbreviation BSCVA in lines 31-32 and 126.

Reply: Thank you for pointing this out. Indeed, correction was occasionally conducted with contact lens rather than with spectacles. We have changed “BSCVA” to “BCVA” throughout the manuscript.

14) Line 279: The authors might would like to specify whether the rise of regrafts was attributable to a specific keratoplasty technique.

Reply: The increase in regraft cases is indeed due to the advent of both DSAEK and DMEK. When only PKP could be offered, surgery was conducted as late as possible because the surgery is more invasive and the expected results were relatively limited. By contrast, because DSAEK and DMEK are less invasive, they can be offered earlier, when the patient is less likely to have severe disease and the outcomes are much better. However, because these grafts are extremely fragile, they fail significantly more often than PKP grafts and must be replaced with another graft; depending on the case, it could be a DMEK, DSAEK, or PKP graft. To address this comment, the following texts have been added:

Introduction: “This so-called Descemet stripping automated endothelial keratoplasty (DSAEK) procedure rapidly started replacing PKP worldwide because its lower invasiveness and complication rates meant that it could be offered to the patient before the disease had advanced; this resulted in significantly superior outcomes [14].” Page 4, lines 87–90

Results: “Over the 21 years, 221 keratoplasties were conducted for failure of a first transplant (20%; this includes a 10% total graft rejection and 10% total graft failure rate). The regraft cases started to emerge in 2003 but remained low (1–9/year; mean, 6/year) until 2012. At that point, revision keratoplasties almost tripled. These numbers generally remained high thereafter (mean, 19/year). The increase in regraft cases directly reflects the advent of DSAEK and DMEK. PKP was responsible for nearly all of the graft rejections in the cohort (due to its thick and therefore more immunologically provocative graft) but only 1% of the graft failures. PKP rejections and failures were initially treated with a second PKP until 2012, which was when DSAEK was first used for revision keratoplasty. Thereafter, DSAEK was increasingly used after PKP rejection/failure. By contrast, while the DSAEK and especially DMEK grafts were rarely rejected (due to their thinness), 16% and 11% of these grafts failed, respectively. The DSAEK failures were mostly caused by excessive stromal damage that could not be improved sufficiently with DSAEK. Most were repaired with PKP. The DMEK failures were mostly caused by graft detachment that significantly affected graft function and could not be reversed with rebubbling. These cases were generally reoperated with a second DMEK or a DSAEK. Thus, DSAEK was increasingly performed for regraft; in 2019, it was used for 44% of all revision cases. DMEK was first used for revision keratoplasty in 2014 and its use in regrafts rose modestly over time to 24%. Nonetheless, throughout these changes, PKP remained a favored option for revision cases (Figure 6).” Page 9–10, lines 214–232

15) Line 289: I would suggest using “further progression“ instead of “further degeneration“ in the context of corneal crosslinking treatment to stabilize keratoconus.

Reply: The change has been made. Page 12, line 300

16) The literature review is well written.

Reply: Thank you.

Reviewer #2

The authors present a fascinating article assessing the evolution of types of corneal transplants. The duration of the analysis is 21 years, from 2000 to 2020. The treatments were performed in a clinic specializing in corneal transplants in France. This is a retrospective cohort study. The authors assess indication, surgical techniques, demographic data, and BCVA 6 months after surgery. The total number of patients included in the analysis was 1,042 in the last 20 years.{For the first ten years, only Penetrating grafts have been performed for the 10 first years of analyzing, and then lamellar grafts have started. New procedures have launched in 2011 ( DMEK, DSAEK, DALK). The main indication for keratoplasty changed from Keratoconus to pseudophakic bullous keratopathy (PBK) and all Fuchs endothelial cell dystrophy (FECD). The proportion of penetrating to lamellar corneal grafts in the last year was 27 to 78 percent. The Authors observed improvement of BCVA in all cases. In conclusion, the authors state that the introduction of lamellar grafts to corneal surgery was essential in the change in the treatment of corneal disease

Reply: Thank you very much for your thorough review of our paper and pertinent comments. We have addressed all points to the best of our ability and believe the revisions have improved our paper. . Please note that underlines indicate changed texts in the cited texts below.

The authors are asked to clarify some issues.

Title line 3 : explain the word,, specialized “ is it Corneal Transplant Unit or Corneal Transplant Center ? And change the title.

Reply: We changed the title to:

“Evolution of Corneal Transplantation Techniques and Their Indications in a French Corneal Transplant Unit in 2000–2020” Page 1, lines 2–4

Abstract:

Line 43 ,, age dropped steadily over time.” Age of whom?

Reply: The line was changed as follows:

“Patient age dropped steadily over time.” Page 2, line 43

In the abstract is a lack of numbers ( BCVA before and after),

Reply: To keep the Abstract word count to ≤250 words, we altered the sentence as follows:

“On average, BSCVA improved by 0.1–0.3 logMAR.” Page 2, lines 42–43

Line 49 key words : look for more appropriate ones ( indication, surgical technique)

Reply: Since up to 50 keywords are permitted, we added the full names of the indications and surgical techniques and included DALK and evolution of surgical techniques, as follows:

“Corneal transplantation, evolution of surgical techniques, Descemet membrane endothelial keratoplasty, DMEK, Descemet stripping automated endothelial keratoplasty, DSAEK, penetrating keratoplasty, deep anterior lamellar keratoplasty, DALK, Fuchs endothelial corneal dystrophy, FECD, pseudophakic bullous keratopathy, PBK, keratoconus.” Page 2, lines 49–52

Introduction: line 56 when Eduard Konrad Zirm put ,, in Olomouc in Central Europe. “

Reply: The sentence was adapted as follows:

“This surgery has a history spanning several centuries but first became clinically useful in 1905, when Eduard Konrad Zirm in Olomouc in Central Europe conducted the first successful full-thickness human corneal transplantation.” Page 3, lines 57–59

Line 72 by Anwar and Teichmann in 2002 insert and published in 2002 y. ( They invented this technique earlier before 2000)

Line 74 dissected separated

Reply to both comments: Thank you, the sentence was altered as follows:

“At present, the most commonly used DALK technique is the one that was published by Anwar and Teichmann in 2002, where the stroma is dissected and separated by injecting an air bubble between the pathological stroma and the healthy Descemet membrane.” Page 3, lines 69–72

Line 75-76 The extensive bubble technique is described above, and DELK should be described here, invented by Gerit Melles.

Reply: The original text described the DLEK procedure. To make the text easier to read, it was rewritten as follows:

“in 1998–2000, Melles proposed a method for replacing the diseased endothelium in corneal diseases such as bullous keratopathy and Fuchs endothelial corneal dystrophy (FECD). This method was termed deep lamellar endothelial keratoplasty (DLEK) and involves (i) dissecting a posterior lamellar disc from the recipient cornea, (ii) inserting a folded donor posterior corneal disc consisting of posterior stroma, Descemet membrane, and endothelium into a self-healing 5-mm tunnel incision, (iii) unfolding the disc, and (iv) then securing it against the recipient tissue with an air bubble [7–9].” Page 3–4, lines 74–80

Line 106 insert Aim of the study:

Reply: The Study Aim was clarified as follows:

“The present retrospective study analyzed the cases of all consecutive patients who underwent surgery between 2000 and 2020 in a French corneal transplantation unit to elucidate how the inventions of DALK, DSAEK, and DMEK have affected the surgical landscape, the main indications for corneal transplantation, and the surgical outcomes.” Page 5, lines 105–108

Line 115 This is unclear to me. The study is retrospective. In that case, did the patient know what he agreed to? Was such consent then necessary?

Ethics Committee of the French Society of Ophthalmology (Approval No. 00008855). Is it a number of approval for this study or all these kinds of studies in France?

Reply: The possibility that a patient’s surgery-related data may be used for research purposes is part of the consent form that the patient signs to agree to surgery. However, to avoid confusion, we changed the text. The number is actually the number of the Ethics Committee. The text has been corrected as follows:

“All procedures were conducted in accordance with the principles of the Declaration of Helsinki. The study was approved by the Ethics Committee of the French Society of Ophthalmology (IRB 00008855 Société Française d’Ophtalmologie IRB#1). The need for explicit consent from the patients to use their anonymized surgery-related data was waived due to the retrospective nature of the study.” Page 5, lines 113–117

Line 124 Put initials of a surgeon ( author of the article?)

Reply: Jean-Marc Perone conducted all the surgeries. His initials were added. Page 5, line 122

Line 128 The date are incomplete BCVA only from 2010 why?

Reply: These data were not routinely recorded prospectively in the database until 2010. To address this point, the following change was made:

“BCVA data were only available from 2010 onwards because that was when they started to be routinely and prospectively recorded in the medical database.” Page 5, lines 126–128

Line 129 Do you have data of the thickness of the cornea ?

Reply: Unfortunately, the corneal thickness measurements were not recorded in a routine manner.

Line 251 How many graft rejections occurred in this cohort? How many rebuilding after DSAEK and DMEK occurred?

Reply: The total transplant rejection rate for the entire cohort was 10%. The vast majority were PKP cases. The total graft failure rate (defined as requiring reoperation within 2 postoperative years) was 10%. Most were DSAEK (16% failure rate) and DMEK (11%) grafts. PKP associated with a graft failure rate of 1%. These points have been added to the text as follows:

“Over the 21 years, 221 keratoplasties were conducted for failure of a first transplant (20%; this includes a 10% total graft rejection and 10% total graft failure rate). The regraft cases started to emerge in 2003 but remained low (1–9/year; mean, 6/year) until 2012. At that point, revision keratoplasties almost tripled. These numbers generally remained high thereafter (mean, 19/year). The increase in regraft cases directly reflects the advent of DSAEK and DMEK. PKP was responsible for nearly all of the graft rejections in the cohort (due to its thick and therefore more immunologically provocative graft) but only 1% of the graft failures. PKP rejections and failures were initially treated with a second PKP until 2012, which was when DSAEK was first used for revision keratoplasty. Thereafter, DSAEK was increasingly used after PKP rejection/failure. By contrast, while the DSAEK and especially DMEK grafts were rarely rejected (due to their thinness), 16% and 11% of these grafts failed, respectively. The DSAEK failures were mostly caused by excessive stromal damage that could not be improved sufficiently with DSAEK. Most were repaired with PKP. The DMEK failures were mostly caused by graft detachment that significantly affected graft function and could not be reversed with rebubbling. These cases were generally reoperated with a second DMEK or a DSAEK. Thus, DSAEK was increasingly performed for regraft; in 2019, it was used for 44% of all revision cases. DMEK was first used for revision keratoplasty in 2014 and its use in regrafts rose modestly over time to 24%. Nonetheless, throughout these changes, PKP remained a favored option for revision cases (Figure 6).” Page 9–10, lines 214–232

Line 258 big bubble was not invented in 2002.

Reply: Thank you, we changed the text as follows:

“The publication of air bubble DALK for anterior opacities in 2002 and the invention of first DSAEK for endothelial pathologies in 2004 and then its technically more challenging counterpart DMEK in 2006 has completely revolutionized the field with enormous speed” Page 11, lines 268–271

Discussion

Line 258 not only CXL

Reply: We believe you meant line 285, which stated, “During this time, keratoconus dropped from third to last place as a keratoplasty indication. This is due to the rise of cross-linking (CXL) treatment, which was first used to manage keratoconus in 2003.” We agree that the advent of CXL is not the only reason, improvements in rigid and scleral lenses have also contributed. We added this point to the manuscript as follows:

“During this time, keratoconus dropped from third to last place as a keratoplasty indication. This is partly because of improvements in rigid and scleral lenses but also because of the rise of cross-linking (CXL) treatment” Page 12, lines 295–297

• Please consider to add : 10.1016/j.transproceed.2016.01.056

Reply: Thank you for alerting us to this paper. It has been added to Supplementary Tables S1 and S2 and it is included in the Discussion as Reference 27, as follows:

“A Polish study also recorded a 4.6-fold increase between 1989 and 2014; these changes may reflect legal and infrastructural changes as well as the inventions in lamellar keratoplasty [27].” Page 13, lines 323–325

• Line 350 Please consider to add : 10.3390/jcm10112421

Reply: We cited this paper as Reference 53 and changed the related text slightly, as follows:

“Low uptake of DALK may also reflect the technical difficulties associated with DALK and the fact that PKP may yield slightly better visual outcomes for keratoconus [52,53].” Page 15, lines 362–363

Conclusion

Please add one ending sentence as a conclusion

Reply: We adapted the last line and presented it as a separate paragraph:

“Thus, the inventions of DSAEK and DMEK have enormously advanced the medical field and are greatly improving patient outcomes.” Page 18, lines 439–440

---

## [Decision Letter · Decision Letter 1]

25 Jan 2022

Evolution of Corneal Transplantation Techniques and Their Indications in a French Corneal Transplant Unit in 2000–2020

PONE-D-21-29757R1

Dear Dr. Perone,

We’re pleased to inform you that your manuscript has been judged scientifically suitable for publication and will be formally accepted for publication once it meets all outstanding technical requirements.

Kind regards,

Timo Eppig

Academic Editor

PLOS ONE

Additional Editor Comments (optional):

Reviewers' comments:

Reviewer's Responses to Questions

**Comments to the Author**

1. If the authors have adequately addressed your comments raised in a previous round of review and you feel that this manuscript is now acceptable for publication, you may indicate that here to bypass the “Comments to the Author” section, enter your conflict of interest statement in the “Confidential to Editor” section, and submit your "Accept" recommendation.

Reviewer #1: All comments have been addressed

Reviewer #2: All comments have been addressed

2. Is the manuscript technically sound, and do the data support the conclusions?

Reviewer #1: Yes

Reviewer #2: Yes

3. Has the statistical analysis been performed appropriately and rigorously? 

Reviewer #1: Yes

Reviewer #2: N/A

4. Have the authors made all data underlying the findings in their manuscript fully available?

Reviewer #1: Yes

Reviewer #2: Yes

5. Is the manuscript presented in an intelligible fashion and written in standard English?

Reviewer #1: Yes

Reviewer #2: Yes

6. Review Comments to the Author

Reviewer #1: (No Response)

Reviewer #2: Dear Authors

Thank you very much for for resolving all my comments. I accept this article for publication in PLOS ONE

7. PLOS authors have the option to publish the peer review history of their article (what does this mean?). If published, this will include your full peer review and any attached files.

Reviewer #1: No

Reviewer #2: **Yes: **Edward Wylęgała

---

## [Editor Report · Acceptance letter]

8 Feb 2022

PONE-D-21-29757R1 

Evolution of Corneal Transplantation Techniques and Their Indications in a French Corneal Transplant Unit in 2000–2020  

Dear Dr. Perone:

I'm pleased to inform you that your manuscript has been deemed suitable for publication in PLOS ONE. Congratulations! Your manuscript is now with our production department. 

Kind regards, 

on behalf of

Dr. Timo Eppig 

Academic Editor

PLOS ONE